DOI: 10.1038/s41467-018-05829-7　　OPEN

# Overexpressing lncRNA *LAIR* increases grain yield and regulates neighbouring gene cluster expression in rice

Ying Wang[1,2], Xiaojin Luo[1], Fan Sun[1], Jianhua Hu[2,3], Xiaojun Zha[4], Wei Su[1,5] & Jinshui Yang[1]

Long noncoding RNAs (lncRNAs) are essential regulators of gene expression in eukaryotes. Despite increasing knowledge on the function of lncRNAs, little is known about their effects on crop yield. Here, we identify a lncRNA transcribed from the antisense strand of neighbouring gene *LRK* (leucine-rich repeat receptor kinase) cluster named *LAIR* (*LRK* Antisense Intergenic RNA). *LAIR* overexpression increases rice grain yield and upregulates the expression of several *LRK* genes. Additionally, chromatin immunoprecipitation assay results indicate H3K4me3 and H4K16ac are significantly enriched at the activated *LRK1* genomic region. *LAIR* binds histone modification proteins OsMOF and OsWDR5 in rice cells, which are enriched in *LRK1* gene region. Moreover, *LAIR* is demonstrated to bind 5′ and 3′ untranslated regions of *LRK1* gene. Overall, this study reveals the role of lncRNA *LAIR* in regulating rice grain yield and lncRNAs may be useful targets for crop breeding.

[1] State Key Laboratory of Genetic Engineering, Institute of Genetics, School of Life Sciences, Fudan University, Shanghai 200438, China. [2] Research Center for Ecological Science and Technology, Fudan Zhangjiang Institute, Shanghai 201203, China. [3] State Key Laboratory of Molecular Engineering of Polymers, Department of Macromolecular Science, Fudan University, Shanghai 200433, China. [4] College of Chemistry and Life Science, Zhejiang Normal University, Jinhua 321004, China. [5] State Key Laboratory of Genetic Engineering, Institute of Plant Biology, Department of Biochemistry, School of Life Sciences, Fudan University, Shanghai 200438, China. Correspondence and requests for materials should be addressed to W.S. (email: weisu@fudan.edu.cn) or to J.Y. (email: jsyang@fudan.edu.cn)

Rice (*Oryza sativa L.*) is a staple food crop for more than 3 billion people worldwide[1]. It is also considered as one of the most important model monocotyledonous plants for genome research due to its small genome size and considerable synteny with other cereal genomes[2,3]. Yield is a crucial agronomic trait for cereal crops' genetic improvement. Its improvement relies on the manipulation of a range of component traits such as number of panicles, grains per plant, and 1000-grain weight, which often exhibit continuous phenotypic variations. It has been shown that these traits are regulated by quantitative trait loci (QTLs), whose effects are usually mediated by at least two genes and environmental conditions.

The traditional definition of a gene has recently been challenged by the discovery of noncoding RNAs (ncRNAs), which include functional transcripts encoded by a specific genome region. A large-scale sequencing analysis reveals that most of the eukaryotic genome is transcribed[4], and large repertoires of RNAs are generated, including short (22–33 nucleotides) and long (>200 nucleotides) ncRNAs. Long noncoding RNAs (lncRNAs), which lack any obvious open reading frames (ORFs)[5–7], function as key regulators of diverse mechanisms in a range of biological processes[6,8,9]. They may regulate the expression of neighbouring genes with *cis*-acting (close to the transcription site) or *trans*-acting (multiple loci throughout the genome) mechanisms[10]. Approximately one-third of lncRNAs can interact with multiple protein partners[11]. The concept of lncRNAs serving as molecular scaffolds that help assemble and target the chromatin-modifying complex to specific genomic loci for the regulation of gene expression is likely part of a globally emerging mechanistic theme[6,12]. One major theme involves the regulatory role of lncRNAs regarding the expression of nearby protein-coding genes or gene clusters. For example, HOX transcript antisense RNA (*HOTAIR*), one of the first identified *trans*-acting lncRNAs, binds to multiple genomic loci of *HOXD* genes and serves as a molecular scaffold that enables the recruitment of chromatin-modifying complexes to repress the expression of *HOXD* genes[13,14]. In *Drosophila melanogaster*, the dosage compensation complex with lncRNAs [RNA on X1 (*roX1*) and RNA on X2 (*roX2*)] that combine with proteins [male-specific lethal-1 (MSL-1), MSL-2, MSL-3, Males-absent-on-the-first (MOF), and Maleless (MLE)] acetylates H4K16 so that chromatin can assume a more open conformation in the hyperactive cells of male fruit flies[15–17]. Genomic occupancy maps of *roX* RNAs have also been done to show the targeting of X chromosomes[18].

In plants, it is known that lncRNAs can regulate gene silencing, flowering time, reproduction, stress responses, organogenesis in roots, and photomorphogenesis in seedlings[19–24]. Because the overwhelming majority of well-characterized plant lncRNAs with established functions were studied in *Arabidopsis thaliana*, our understanding of the mechanisms regulating lncRNAs in crop species remains limited. Thus, elucidating the mechanisms of action for the rice lncRNAs may clarify the functions of ncRNAs in different species and provide useful information for the breeding of food crops. We previously detected and cloned a leucine-rich repeat receptor kinase (LRK) gene cluster from a QTL associated with increased grain yield in rice[25]. In this study, we identify a lncRNA transcript (*LRK* Antisense Intergenic RNA, *LAIR*) from the antisense strand of the *LRK* gene cluster. Transgenic lines, in which *LAIR* is silenced, exhibit inhibited growth and low expression levels for all *LRK* genes. In contrast, considerable increases in total grain yield and upregulated expression of some members of the *LRK* gene cluster are observed in lines overexpressing *LAIR*. Moreover, luciferase reporter assays reveal that *LAIR* is capable of differentially activating promoters of the *LRKs* gene. Meanwhile, an analysis of chromatin modifications related to *LRK* genes indicates H3K4me3 and H4K16ac

are enriched in transcriptionally active genes. Next, *LAIR* binds histone modification proteins OsMOF and OsWDR5 (WD repeat domain 5) in rice cells, which are detected enrichment in the *LRK1* gene region. Additionally, ChIRP assays reveal that *LAIR* binds 5′ and 3′ UTR genomic regions of the *LRK1* gene. These results imply that *LAIR* overexpression increases rice yield and regulates the expression of neighbouring *LRK* gene cluster.

## Results

**Identification of the antisense lncRNA *LAIR*.** We previously identified a QTL for improved rice yield consisting of several tandemly arranged intronless *LRK* gene copies in a 51.3-kb region of chromosome 2[25]. The *LRK* gene cluster in the *O. sativa* ssp. *indica* MH63 genome was subsequently sequenced. Genomic and transcriptomic analyses revealed the *LRK* gene cluster in MH63 contained eight highly similar *LRK* genes (*LRK1–8*), which were transcribed normally, with the exception of *LRK5* (Fig. 1a).

An antisense transcript encoded by the 5′ terminal region of the *LRK* gene cluster was named *LAIR*. The *LRK1* gene was located in the first intron of *LAIR*, and there was no overlap between the two antisense transcripts (Fig. 1a). The 5′ and 3′ ends of *LAIR* were determined by rapid amplification of cDNA ends (RACE) (Supplementary Fig. 3). We observed that *LAIR* contains a 5′ cap structure and a polyadenylated 3′ end, which is typical of mRNAs. Thus, it is likely that RNA polymerase II (RNAPII) is responsible for the transcription of *LAIR*. The spatial and temporal expression of *LAIR* was analyzed, *LAIR* showed relatively high levels in 3-leaf-stage shoot, flowering-stage node, pistil and caryopsis (Supplementary Fig. 1). Additionally, up to 10 alternative spliced isoforms of *LAIR* were identified and sequenced (JX512719–JX512728). All splicing events were detected in the second exon of *LAIR*, and consisted of the shuffling of specific motifs rather than random events. The most enriched *LAIR* RNA (JX512726) was 1797 bp long, while the shortest *LAIR* RNA (JX512719) was 1585 bp long. The corresponding genomic and transcript sequences were up to 7.3 kb and approximately 1.8 kb long, respectively (Supplementary Fig. 2 and Fig. 1a).

We evaluated the coding potential of *LAIR*. The ORF Finder online tool (https://www.ncbi.nlm.nih.gov/orffinder/) was used to predict ORFs based on *LAIR* isoform JX512726. As indicated in Fig. 2a, the termination codon was distributed in all three frames, and only a few sequences encoding short peptides were detected. The two longest peptides comprised 133 and 104 amino acids. Furthermore, there were no functional domain matches for any of the peptides according to homology searches of large databases of protein families and domains (Pfam[26] and SMART[27]). The possibility that these peptides function as small proteins was assessed by the following genetic investigations. An analysis of the protein-coding potential of transcripts[28] indicated that all 10 alternative spliced isoforms of *LAIR* represented ncRNAs (Fig. 2b). Additionally, the DNA sequences of the different alternatively spliced *LAIR* isoforms had diverse coding frames. Therefore, we hypothesized that there is no functional constraint associated with the *LAIR* coding regions.

To summarize, we identified an antisense lncRNA (*LAIR*) from the 5′ terminal region of the *LRK* gene cluster; further analyses indicated its association with rice grain yield.

**LAIR affects rice yield traits.** To test whether *LAIR* is functional in individual plants, we conducted genetic transformation experiments. Full-length cDNA sequences of sense or antisense *LAIR* (JX512726) were cloned and transformed into rice MH63 under the control of the 35S Cauliflower Mosaic Virus (CaMV) promoter (35S::LAIR and 35S::anti-LAIR). Compared with the

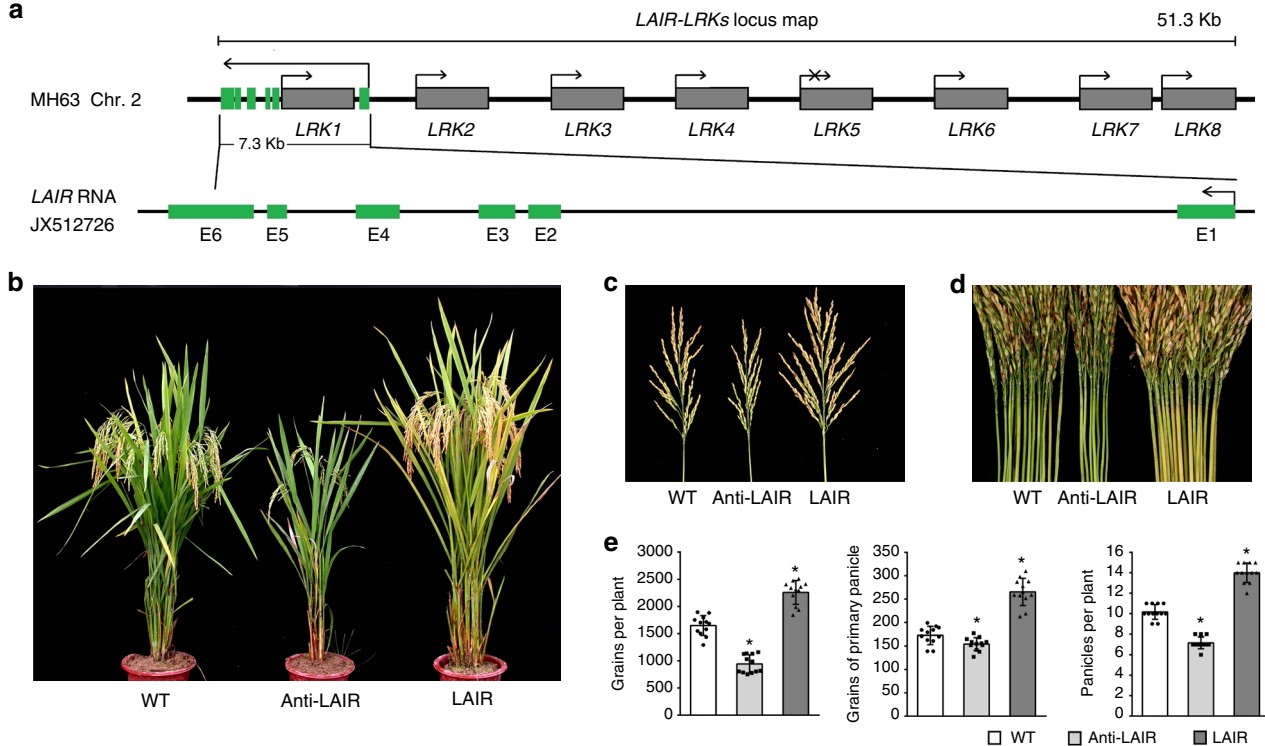

**Fig. 1** Cloning and characterization of *LAIR*. **a** Genome map of the *LRK* gene cluster and *LAIR*. The *LRK* genes are located in a 51.3-kb region on chromosome 2 of Nipponbare PAC clone AP007224. LncRNA *LAIR* (JX512726), which includes six exons, is transcribed from the antisense strand of *LRK1*. Additionally, *LRK1* is in the first *LAIR* intron, with no overlap between the two transcripts. **b**–**d** Phenotypes of wild-type (WT), antisense RNAi-*LAIR* (Anti-LAIR), and *LAIR*-overexpressing (LAIR) transgenic rice plants. Growth traits of $T_3$ transgenic lines were investigated, including mature plant phenotype (**b**), primary panicle size (**c**), and number of panicles per plant (**d**). **e** Growth traits of Anti-LAIR and LAIR transgenic lines. Student's *t*-test: *$P < 0.05$. Data are presented as the mean ± standard deviation ($n = 12$)

wild-type control plants, the 35S::LAIR plants had larger primary panicles and more panicles per plant, leading to a considerable increase in total grain yield per plant (Fig. 1b–e). In contrast, the 35S::anti-LAIR transgenic plants exhibited inhibited growth and decreased grain yield. Similar rice growth traits were reported for *LRK1*-overexpressing transgenic lines[29]. These results demonstrate that *LAIR* expression affects rice plant growth and grain yield.

The *LAIR* transcript is long (approximately 1.8 kb) and was predicted to lack coding potential according to the ORF and functional domain analyses. To investigate whether *LAIR* functions as a lncRNA, we mutated the sequences encoding the two longest peptides (133 and 104 amino acids according to the JX512726 sequence). Two *LAIR* mutant vectors were constructed, with a single nucleotide mutation introducing a premature stop codon in the LAIR-MU1 mutant, and a single deleted nucleotide after the start codon (ATG) in the LAIR-MU2 mutant (Fig. 2c). We analyzed transgenic rice MH63 plants carrying the 35S::LAIR-MU1 and 35S::LAIR-MU2 constructs and observed that the morphological characteristics and grain yield of these mutants *LAIR* lines were similar to those of the wild-type *LAIR*-overexpressing (35S::LAIR) transgenic lines (Fig. 2d–f). These results implied that *LAIR* functions were not dependent on the encoded products. Thus, *LAIR* likely serves as a lncRNA that regulates rice plant growth and increases grain yield.

**LAIR influences the transcription of *LRK* gene cluster.** An earlier study that confirmed non-coding loci can affect the expression of neighbouring protein-coding genes was critical for

the functional characterization of ncRNAs[30]. Additionally, the 35S::LAIR transgenic rice lines exhibited improved growth traits that were consistent with the growth characteristics of the *LRK1*-overexpressing transgenic rice lines. Thus, a qRT-PCR analysis was used to investigate the expression of *LRK* genes in *LAIR*-associated transgenic rice lines. In the *LAIR*-overexpressing 35S::LAIR lines, the *LRK* genes were differentially expressed. The *LRK1* and *LRK4* transcript levels were high, while the expression levels for the other *LRK* genes remained unchanged or decreased slightly (Fig. 3a). Furthermore, *LRK* genes expression in the *LAIR* mutant lines with the 35S::LAIR-MU1 or 35S::LAIR-MU2 constructs was similar to that in the 35S::LAIR lines (Fig. 3b). Accordingly, *LAIR* may have a key role influencing the transcriptional regulation of *LRK* genes. All *LRK* genes were expressed at relatively low levels in the 35S::anti-LAIR lines, suggesting that *LAIR* was crucial for the expression of the *LRK* gene cluster (Fig. 3d).

To clarify the regulatory effects of *LAIR* on the expression of *LRK* genes, rice protoplasts were transformed with sense or antisense *LAIR* expression vectors along with the vector harboring the *LRK* promoter-driven luciferase reporter gene. The LUC/REN signal ratios suggested that wild-type *LAIR* (LAIR) could significantly promote the promoter activations of *LRK1* and *LRK4*. Simultaneously, mutant *LAIR* (LAIR-MU1) showed similar activity ability on *LRKs* (Fig. 3c). On the contrary, antisense *LAIR* (anti-LAIR) suppressed the promoter activations of all the *LRKs* (Fig. 3e). The different regulatory effects between sense *LAIR* and antisense *LAIR* may be caused by the fact that sense *LAIR* transformation only contained one isoform (JX512726) of alternative spliced *LAIR*, and antisense *LAIR* could

interfere all isoforms. Together, our results implied that *LAIR* was capable of activating the *LRK* promoter in vivo, with different regulatory effects among different *LRK* loci.

**LAIR overexpression changes histone modification marks**. Recent advances revealed the broad functional repertoire of lncRNAs, including their effects on chromatin modifications and epigenetic changes to specific genomic loci that regulate the expression of neighbouring protein-coding genes[30]. To assess whether altered *LAIR* expression results in changes of histone modifications, we completed a chromatin immunoprecipitation (ChIP) assay to investigate the modification status of *LRK* loci. We analyzed the histone modifications associated with transcriptional activation, including trimethylated histone H3 lysine 4 (H3K4me3), acetylated H4K5 (H4K5ac), H3K9ac, H4K16ac, and H3K27ac[31–34]. We mainly focused on *LRK1* because its transcription was increased in the 35S::LAIR lines, while *LRK3* was used as a control because its transcription was unaffected by *LAIR* expression (Fig. 3a–c).

Our ChIP–qRT-PCR results indicated that H3K4me3 and H4K16ac were unaffected at the *LRK3* locus, but were obviously enriched at the promoter, 3′ untranslated region, and *LRK1* gene body in the *LAIR*-overexpressing line relative to the levels in the wild-type plants (Fig. 4b). These observations suggested that the overexpression of *LAIR* increased the H3K4me3 and H4K16ac levels at the *LRK1* chromatin region, which was consistent with

the upregulated transcription of *LRK1* in the 35S::LAIR lines. In contrast, when enrichments of H3K4me3 and H4K16ac were analyzed in the 35S::anti-LAIR lines, reduction were observed both at the *LRK1* and *LRK3* locus (Fig. 4c), which were consistent with the suppression of anti-LAIR to *LRKs* promoter (Fig. 3d, e). Additionally, analysis of histone modifications on *LRK4* which was also activated by *LAIR* (Fig. 3a–c) was completed. The H3K4me3 and H4K16ac levels were increased in the 35S::LAIR lines, and reduced in the 35S::anti-LAIR lines (Supplementary Fig. 5). The results could exclude the possibility that the changes of H3K4me3 and H4K16ac levels at *LRK1* locus might be caused by the transcription of *LAIR* itself. Taken together, the results suggested that *LAIR* overexpression resulted in altered expression of specific *LRK* genes and correlated with the changes in histone modifications.

**LAIR binds OsMOF and OsWDR5 that target LRK1 genomic region**. A considerable proportion of lncRNAs was recently confirmed to associate with chromatin-modifying complexes and modulate gene transcription[11]. Because of the histone modification changes of H3K4me3 and H4K16ac in our ChIP assay (Fig. 4b, c), we tested the interaction between *LAIR* and chromatin-modifying complex proteins. The MOF enzyme is the primary histone H4K16 acetyltransferase in mammalian cells[35], and is reportedly a component of MSL and the nonspecific lethal (NSL) complex[36,37]. An increased abundance of H3K4me3 was also identified in the

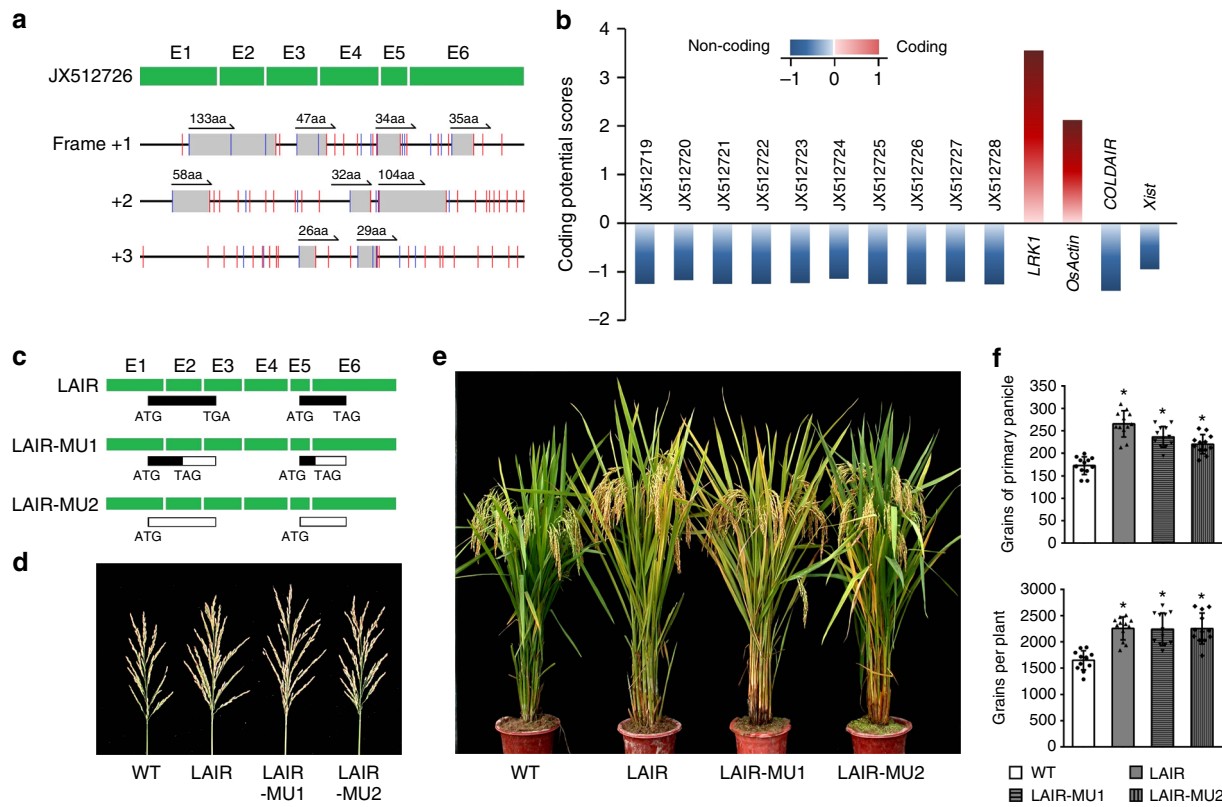

**Fig. 2** *LAIR* coding potential analysis. **a** Analysis of the *LAIR* open reading frame (JX512726). All frames (gray boxes) were identified in the three forward frames. The two longest open reading frames encoded 133 and 104 amino acids (aa). Blue and red lines represent the start and stop codons, respectively. **b** Analysis of the coding potential of the alternatively spliced *LAIR* isoforms (Supplementary Fig. 2). Coding potential scores were generated using the CPC program. Transcripts with scores beyond −1 and 1 are marked as non-coding or coding in this CPC classification[28]. *LRK1* and *OsActin* are provided as coding examples, while *COLDAIR* and *Xist* represent non-coding examples. **c** *LAIR*-mutant vectors were constructed based on the two longest open reading frames, which encoded 133 aa and 104 aa. A single nucleotide mutation introduced a premature stop codon in LAIR-MU1. A single nucleotide was deleted after the start codon (ATG) in LAIR-MU2. **d**, **e** Phenotypes of WT, LAIR, and transgenic plants carrying the *LAIR* mutant vectors. Growth traits of T₃ transgenic lines were investigated, including mature plant phenotype (**e**) and primary panicle size (**d**). **f** Growth traits of LAIR-MU1 and LAIR-MU2 transgenic lines. Student's *t*-test: *$P < 0.05$. Data are presented as the mean ± standard deviation ($n = 12$)

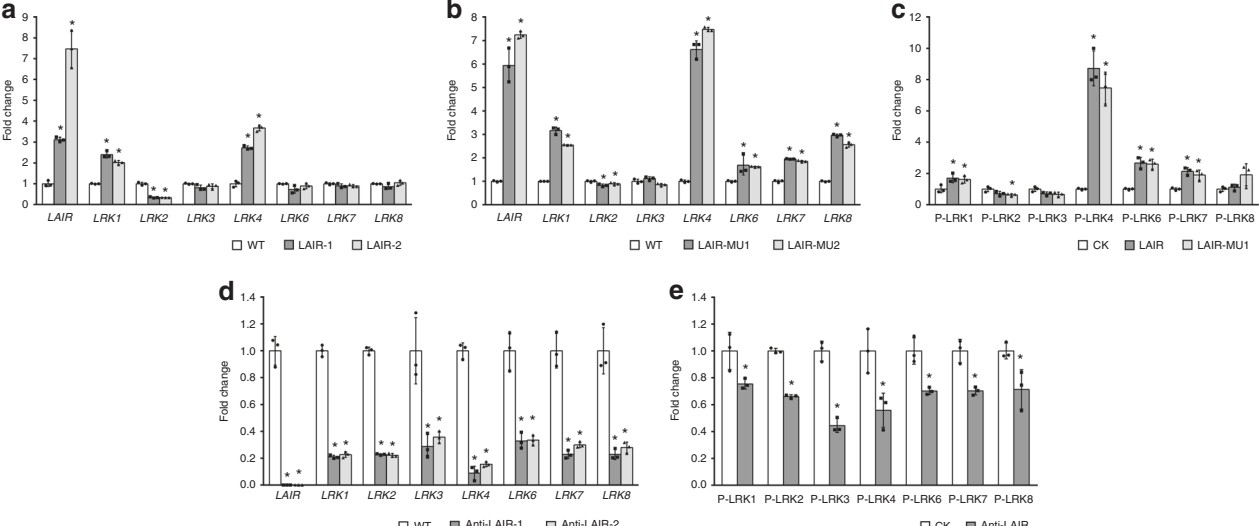

**Fig. 3** Differential regulation of *LRK* gene expression by *LAIR*. **a**, **b**, **d** qRT-PCR analysis of *LRK* expression in transgenic rice lines. *LAIR*-overexpressing (**a**, LAIR-1 and LAIR-2), *LAIR*-mutant (**b**, LAIR-MU1 and LAIR-MU2), and antisense RNAi-*LAIR* vectors (**d**, Anti-LAIR-1 and Anti-LAIR-2). WT: wild-type rice. *OsActin* was used as a reference control, and gene expression levels in WT lines were used for data normalization. **c**, **e** Regulation of the *LRK* gene promoter by sense or antisense *LAIR*. Relative LUC activity from the transient expression of the *LRK* promoter-driven luciferase reporter gene co-infiltrated with sense-*LAIR* (**c**, LAIR), *LAIR*-mutant (**c**, LAIR-MU1), or antisense *LAIR* (**e**, Anti-LAIR) vectors. CK: empty vector control. The CK LUC/REN signal ratios were used for data normalization. All data are presented as the mean ± standard deviation ($n = 3$). Student's *t*-test: *$P < 0.05$

promoter regions of NSL-targeted genes[38]. Additionally, MLE is a critical protein that associates with ncRNAs to form the MSL complex[39,40]. An NSL component, WDR5, has recently been reported to interact with hundreds of lncRNAs[33,41].

Homologs of human *MOF* (NM_032188), *D. melanogaster MLE* (NM_057293), and human *WDR5* (NM_017588) in rice (*OsMOF* XM_015789227, *OsMLE* XM_015756470, and *OsWDR5* XM_015772971, respectively) were cloned and fused in-frame with the FLAG tag sequence. Rice protoplasts were co-infiltrated with these genes and the 35S::LAIR vector or 35S::anti-LAIR for a RNA immunoprecipitation (RIP) assay. Our RIP–qRT-PCR results revealed the significant binding of OsMOF and OsWDR5, but not OsMLE, to the 5′ and 3′ regions of *LAIR* (Fig. 4d). These results were further confirmed by RIP assay using the OsMOF-FLAG and OsWDR5-FLAG transgenic rice lines (Supplementary Fig. 4). Thus, *LAIR* appears to be able to interact with two subunits of the chromatin-modifying complex in rice cells.

Furthermore, to investigate the binding ability of OsMOF and OsWDR5 to *LRKs* genomic loci, we performed ChIP assay in the OsMOF-FLAG and OsWDR5-FLAG transgenic rice lines. A notable enrichment was detected in the *LRK1* region, while the binding levels at the *LRK3* locus were similar to those of the controls (Fig. 4e). Our findings confirmed that OsMOF and OsWDR5 can target *LRK1* genomic region.

**_LAIR_ binds to _LRK_ genomic loci**. To investigate the association between genomic DNA region of *LRKs* and *LAIR*, we performed chromatin isolation by RNA purification (ChIRP) assay. Successful enrichment of *LAIR* RNA using biotinylated LAIR-antisense DNA (LAIR-asDNA) probes was confirmed by *LAIR* 5′-primer qRT-PCR. The result showed that LAIR-asDNA probes retrieved ~65% of *LAIR* RNA and undetectable *OsActin* (Fig. 4f, left panel). Identification of DNA fraction extracted from ChIRP samples showed that *LAIR* significantly occupied at the 5′ and 3′ untranslated regions of *LRK1* compared with *LRK3* (Fig. 4f, right panel). These data suggested that *LAIR* RNA could physically locate to *LRKs* gene genomic DNA region, with different effects among the *LRK* loci.

## Discussion

There has recently been considerable interest in the noncoding transcripts in diverse eukaryotes. Accordingly, there has been a rapid increase in the number of well-studied lncRNAs in mammals, while related plant studies have lagged behind. In this study, a lncRNA, *LAIR*, was identified in rice. Most of the tens of thousands of ncRNA sequences encoded in plant genomes are transcribed by RNAPII, and the majority of the identified plant lncRNAs are polyadenylated[42]. Similarly, *LAIR* contains a 5′ cap structure and a polyadenylated 3′ end, implying RNAPII is responsible for the transcription of *LAIR* as well as the nearby protein-coding genes. Moreover, it is likely that most of the uncharacterized lncRNAs help regulate the expression of neighbouring genes[43]. The *LAIR* sequence is encoded in the antisense strand of the 5′ terminal region of the *LRK* gene cluster. Additionally, *LRK1* is located in the first intron of *LAIR*, with no overlap between the two antisense transcripts (Fig. 1a). Thus, *LAIR* functions may not involve a direct base-paired interaction with the transcript of the protein-coding *LRK* genes. Furthermore, we discovered that *LAIR* overexpression can result in highly upregulated expression of several *LRK* genes as well as an obvious increase in rice grain yield. Meanwhile, the increased transcription of antisense *LAIR* downregulates the expression of all *LRK* genes and inhibits plant growth (Figs. 1b–e and 3a, d). Considering exogenous *LAIR* was randomly incorporated into the rice genome and not near the *LRK* gene cluster, we concluded that *LAIR* is a crucial *trans*-acting regulator of *LRK* transcription. We also detected an intriguing gene expression pattern in *LAIR*-over-expressing rice lines. Specifically, there were differences in the expression level changes among the *LRK* genes (Fig. 3a). These differences were also observed in the luciferase reporter assays (Fig. 3c). Therefore, *LAIR* regulates *LRKs* expression with diverse effects among genes.

The mechanisms underlying the lncRNA-associated regulation of gene expression are very diverse. Recent reports described a crucial role for lncRNAs involving epigenetic modifications to specific genomic loci, which then regulate the expression of neighbouring protein-coding genes[6,30,44]. Given that *LAIR* activates the expression of *LRK* genes, we analyzed the histone

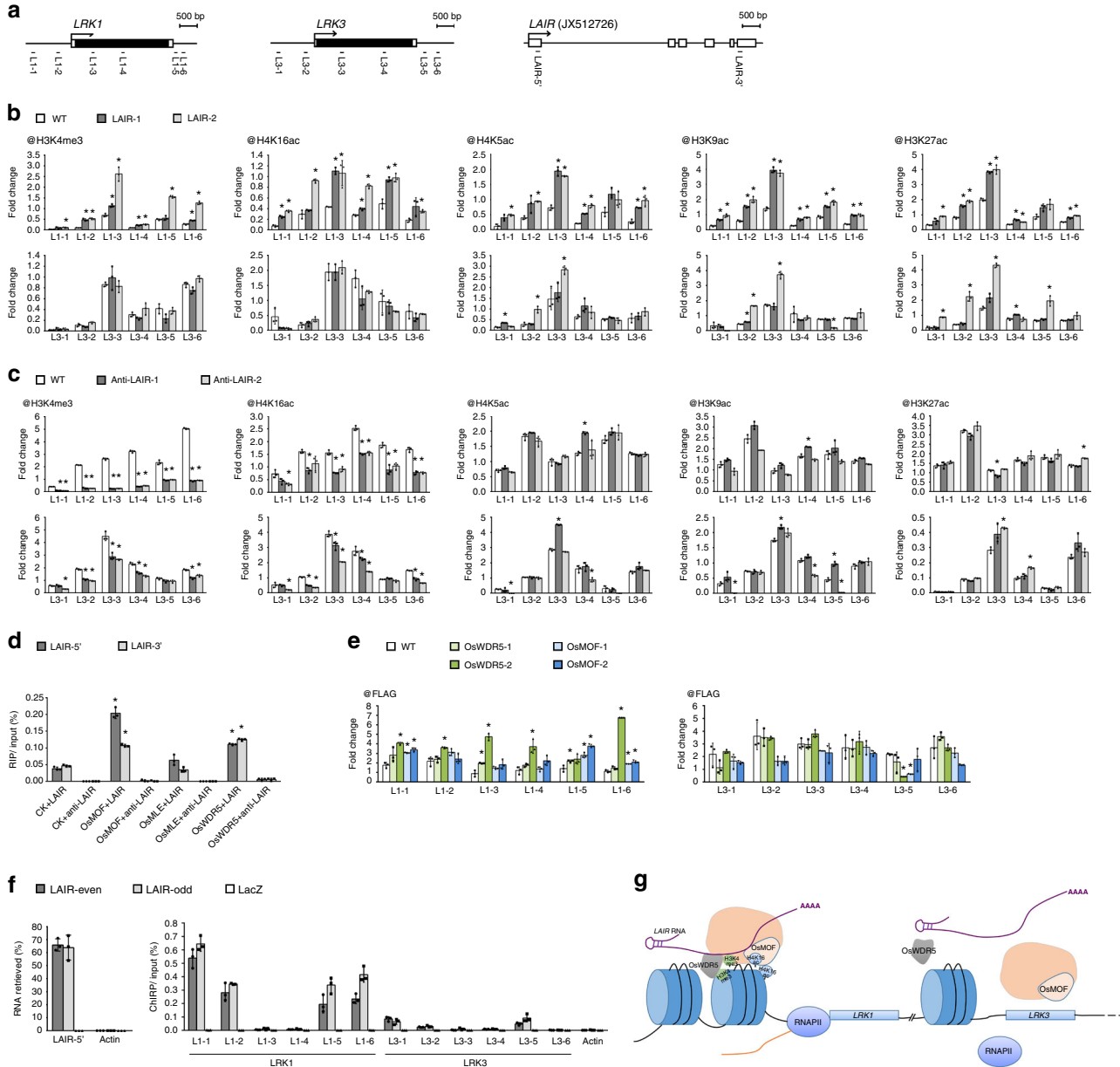

**Fig. 4** Molecular processes involving *LAIR* RNA, epigenetic modification proteins and chromatin of *LRK* genes. **a** Schematic representation of *LRK1, LRK3*, and *LAIR* (JX512726) gene structure and primers position indicating the regions examined in the ChIP, RIP, and ChIRP assays of **b**–**f**. Black boxes indicate open reading frame. **b**, **c** ChIP assays of *LRK* loci. The ChIP analyses were completed for the *LRK1* and *LRK3* chromatin regions using antibodies against H3K4me3, H4K5ac, H3K9ac, H4K16ac, and H3K27ac in rice wild-type (WT) and *LAIR*-overexpressing (LAIR-1 and LAIR-2) lines (**b**) or antisense RNAi-*LAIR* lines (Anti-LAIR-1 and Anti-LAIR-2) (**c**). *OsActin* was used as a reference control and for data normalization. **d** RIP assays of epigenetic modification-associated proteins (OsMOF, OsMLE, and OsWDR5) confirmed the interaction with *LAIR*. The proteins were fused to the FLAG tag. CK: empty vector control with only the FLAG tag. The RIP was completed in rice protoplasts that were co-infiltrated with fusion proteins vector or the control, and the vector with the 35S::LAIR or 35S::anti-LAIR. **e** ChIP assays of modification-associated proteins on *LRK* loci. The analyses were completed for *LRK1* and *LRK3* using an anti-FLAG antibody in rice wild-type (WT) as well as OsWDR5-FLAG (OsWDR5-1 and OsWDR5-2) and OsMOF-FLAG (OsMOF-1 and OsMOF-2) transgenic lines. *OsActin* was used as a reference control and for data normalization. **f** ChIRP assays of *LAIR* on *LRK* loci. ChIRP enriched ~65% of *LAIR* RNA in both odd and even probes pool, with undetectable *OsActin* (Actin) (left panel). Immunoprecipitated DNA genomic region analyzed by ChIRP-qPCR showed the remarkable interaction of *LAIR* with the 5′ and 3′ untranslated regions of *LRK1* compare with *LRK3* (right panel). Probes that targeted *LacZ* mRNA were served as negative controls. **b**–**f** Student's *t*-test: \**P* < 0.05. Data are presented as the mean ± standard deviation (*n* = 3). **g** A proposed model of *LAIR* during the regulation of *LRKs*. *LAIR* interacts with epigenetic regulation proteins OsMOF and OsWDR5; both lncRNA and epigenetic modification proteins locate to *LRK1* genomic region; and *LAIR* overexpression results in changes in histone marks (H3K4me3 and H4K16ac) and activating transcription of *LRK1*. The process is not executed on *LRK3* genomic region. This difference may contribute to the observed differential regulation of *LRK* genes

modification status of *LRK* loci. We observed that H3K4me3 and H4K16ac were enriched at *LRK1* in *LAIR* overexpression lines, implying that epigenetic changes to specific *LRK* loci were associated with altered *LAIR* expression (Fig. 4b). Hundreds of lncRNAs have been confirmed to interact with multiple protein partners, suggesting lncRNAs influence gene expression by targeting chromatin remodelers to specific genomic regions as part of a molecular scaffold[6,11]. RNA-binding proteins OsMOF and OsWDR5, which have shown to associate with the H3K4me3 and H4K16ac histone modification complex, were detected significantly binding with *LAIR* (Fig. 4d and Supplementary Fig. 4). Additionally, OsMOF and OsWDR5 were enriched at the *LRK1* locus (Fig. 4e). Finally, we found evidence on *LAIR* RNA binding with *LRK1* genomic locus (Fig. 4f). Thus, *LAIR* could interact with epigenetic regulation proteins OsMOF and OsWDR5; both lncRNA and epigenetic modification proteins could locate to specific *LRK* genes; and *LAIR* overexpression resulted in altered *LRK* genes expression and changes in histone marks (Fig. 4g). However, current evidence do not support that *LAIR* acts through *LRKs* to regulate grain yield and it is not clear whether the changes in histone marks are the cause of *LRKs* expression or occurring as a consequence.

While the lncRNA activities continue to be clarified, the mechanisms regulating plant lncRNAs remain relatively uncharacterized. As a key histone acetyltransferase in mammalian cells, MOF has been well studied in human and fruit fly, but the research of MOF in rice is almost zero. There are currently no published reports describing lncRNA-regulated gene expression associated with the histone modification protein OsMOF in rice. The NSL complex is a universal and sex-independent major regulator of housekeeping genes[37,45,46]. Because two of its crucial components (OsMOF and OsWDR5) interact with *LAIR*, it is likely that *LAIR* functions are associated with the evolutionarily conserved NSL complex. However, additional research is required to elucidate this potential relationship.

The biological features of lncRNAs have attracted widespread interest, and important advances have been made in nearly all cellular systems or species that have been studied at the genome level. Regarding applied research, current efforts are mostly related to the potential utility of lncRNAs in cancer therapeutics, regenerative medicine involving stem cell technology, and biomarker development for molecular interventions[42,47–50]. However, studies on the application of lncRNAs in plants, especially for crop breeding, remain limited. To satisfy future food and bio-fuel demands, innovative molecular methods to improve crop yields must be developed. The rice *LAIR* described herein may be useful for enhancing plant growth and grain yield. Relative to the wild-type plants, the *LAIR*-overexpressing transgenic rice lines produced larger primary panicles and had more panicles per plant, which likely contributed to the considerable increase in total grain yield (Fig. 1b–e). As a new genetic element identified at a QTL, *LAIR* may represent a new option for introducing stable and heritable increases of grain yield in rice. Rice is considered a model cereal crop because of its well-established genetic background and molecular breeding system. Thus, it may be relevant for characterizing lncRNAs in terms of the associated molecular mechanisms and individual gene expression levels, as well as for the application of lncRNAs in crop breeding. Future studies should aim to characterize the molecular mechanism underlying *LAIR*-regulated gene expression with diverse effects, and explore the potential application of *LAIR* for improving crop yield.

## Methods

**Plant materials and gene transformation**. *O. sativa* ssp. *indica* rice cultivar MingHui63 (MH63) was used in this study. Additionally, the binary vector pCAMBIA1304 (Center for the Application of Molecular Biology to International Agriculture), which carries the kanamycin- and hygromycin-resistance genes for screening transformed bacteria and plants, respectively, was assembled. The *LAIR* isoform of JX512726 was cloned and used to construct 35S:LAIR transgenic line. To induce rice embryogenic calli, dehusked MH63 rice seeds were surface-sterilized by soaking in 70% (v/v) ethanol for 5–10 min and then 0.1% (w/v) HgCl$_2$ for 20–30 min. Seeds were then washed 5–7 times with sterile distilled water and sown on agar-solidified Murashige and Skoog medium. About 20 days later, the calli were bombarded using a gene gun and selected after a 2-week exposure to 50 mg l$^{-1}$ hygromycin. The hygromycin-resistant calli were regenerated and grown (T$_0$ lines). A total of 10–20 independent T$_0$ lines were tested to confirm they were successfully transformed. The hygromycin gene was amplified by a polymerase chain reaction (PCR) with genomic DNA used as the template.

All rice materials were grown in a greenhouse at 28 °C under a 14 h-light/10 h-dark photoperiod or in test fields in Shanghai (31°11′N, 121°29′E) and Sanya (18° 14′N, 109°31′E), China.

**RNA isolation and quantitative real-time PCR**. Rice lines were grown in a greenhouse until reaching the three-leaf stage, after which the RNAprep Pure Plant Kit (Tiangen, Beijing, China) was used to extract total RNA. First-strand cDNA was synthesized using the PrimeScript™ RT Reagent Kit (Takara, Otsu, Japan). A quantitative real-time (qRT)-PCR was conducted using SYBR® Premix Ex Taq™ (Takara) and the CFX96 Real-Time PCR Detection system (Bio-Rad, Hercules, CA, USA). The qRT-PCR program was as follows: 40 cycles of 95 °C for 5 s and 60 °C for 30 s. Melting and standard curves were prepared and analyzed. *OsActin* was used as a reference gene for normalizing transcript levels. Details regarding the primers used in this study are provided in Supplementary Data 1.

**Rapid amplification of cDNA ends**. Total RNA was extracted from wild-type MH63 plants at the three-leaf stage as described above. The 3′ RACE analysis was completed using the 3′-Full RACE Core Set with PrimeScript™ RTase (Takara), while the 5′ RACE analysis was completed with the 5′-Full RACE Kit (Takara). The 5′ and 3′ cDNA fragments were amplified using gene-specific primers (Supplementary Data 1). Additionally, the RACE products were subsequently characterized by PCR or sequencing.

**Rice protoplasts generation and gene expression analysis**. A 1.5-kb sequence upstream of the ATG start codon (−1500 to −1), which corresponded to the *LRK* promoter region, was amplified with the *LRKn* promoter primers (Supplementary Data 1). The amplified fragment was cloned into the pGreenII 0800-LUC[51] dual luciferase (LUC) assay system transient expression vector so that it was followed by the *LUC* reporter gene.

Leaves from rice plants at the three-leaf stage were treated with cellulase and macerozyme to generate protoplasts, which were transiently transformed with the prepared vectors using a 40% polyethylene glycol solution[52]. After a 12–16 h incubation at 28 °C in darkness, the protoplast lysate was collected and tested using the Dual-Luciferase® Reporter Assay System (Promega, Madison, WI, USA) and the Synergy™ 2 Multi-Mode Reader (BioTek, Winooski, VT, USA). The expression levels of the *LUC* reporter gene under the control of the target promoter and the *Renilla* luciferase (*REN*) reporter gene under the control of the CaMV 35S promoter were analyzed. The LUC/REN signal ratio can be used to determine the relative ability of ncRNAs to induce the expression of genes under the control of different promoter sequences.

**Chromatin immunoprecipitation assay**. Two-week-old rice seedlings were harvested and immersed in a fixing buffer [0.4 M sucrose, 10 mM Tris–HCl, pH 8.0, 1 mM EDTA, pH 8.0, 1% formaldehyde, and a protease inhibitor cocktail tablet (Roche, Indianapolis, IN, USA)] under vacuum conditions for 15–20 min. The seedlings were incubated for an additional 5 min after 0.1 M glycine was added. Seedlings were rinsed 5 times with distilled water and then frozen with liquid nitrogen. Fixed seedlings were ground to a powder and resuspended in 30 ml lysis buffer (50 mM HEPES, pH 7.5, 1 mM EDTA, pH 8.0, 150 mM NaCl, 1% Triton X-100, 10% glycerol, 5 mM β-mercaptoethanol, and a protease inhibitor cocktail tablet). Solutions were filtered through a 100-μm nylon mesh and centrifuged at 1500 × g for 20 min. The pelleted chromatin was resuspended and sheared by sonication to produce approximately 200-bp fragments. After a centrifugation at 13,000 × g for 10 min, the supernatants were transferred to new tubes and the appropriate antibodies were added [2 μl (1:600) anti-trimethyl-H3K4 (Cat. No. ab8580), anti-acetyl-H4K5 (Cat. No. ab51997), anti-acetyl-H3K9 (Cat. No. ab10812), anti-acetyl-H4K16 (Cat. No. ab109463), or anti-acetyl-H3K27 (Cat. No. ab4729); Abcam, Cambridge, UK or 3 μl (1:400) monoclonal anti-FLAG M2 (Cat. No. F1804); Sigma-Aldrich, St. Louis, MO, USA]. Samples were rotated overnight, mixed with 40 μl Magnetic Protein A Beads, and incubated for another 1–1.5 h. Beads were washed with low salt, high salt, LiCl, and TE buffers (1 ml each). Immunocomplexes were eluted from the beads with 500 μl elution buffer (1% SDS and 0.1 M NaHCO$_3$, with 20 μl 5 M NaCl in each tube). Additionally, crosslinks were reversed by incubating samples at 65 °C for at least 5–6 h. Residual proteins were degraded by the addition of a solution consisting of 10 mM EDTA, pH 8.0, 40 mM Tris–HCl, pH 8.0, and 2 μl Proteinase K followed by an incubation at 45 °C for 1 h. Moreover, DNA was purified and resuspended in 100 μl sterile distilled water.

A qRT-PCR was conducted to assess the enrichment of immunoprecipitated DNA during the ChIP experiments. Details regarding the gene-specific primers are listed in Supplementary Data 1. *OsActin* was used as an internal standard for normalizing data.

**RNA immunoprecipitation assay.** The full-length coding sequences for genes associated with epigenetic modifications (*OsMOF*, *OsMLE*, and *OsWDR5*) were cloned from rice MH63 cDNA and fused in-frame with the gene encoding the FLAG tag under the control of the 35S CaMV promoter in the pCAMBIA1306 vector. Rice protoplasts were co-infiltrated with this vector or the control (i.e., empty vector encoding the FLAG tag) and the vector with the 35S::LAIR or 35S:: anti-LAIR vector. After a 12–16-h incubation at 28 °C in darkness, protoplasts were harvested for RIP assays.

The RIP assays were conducted using the EZ-Magna RIP™ Kit (Millipore, Bedford, MA, USA). Monoclonal anti-FLAG M2 (Cat. No. F1804, Sigma-Aldrich, St. Louis, MO, USA) was added (5 μl) to each reaction. Purified RNA was used as the template to prepare first-strand cDNA with the PrimeScript™ RT Reagent Kit. A qRT-PCR was completed using gene-specific primers (Supplementary Data 1) to assess the enrichment of the immunoprecipitated RNA, with *OsActin* serving as the reference control. The RIP efficiency of the protoplasts transformed with the control vector was used to normalize data.

**Chromatin isolation by RNA purification.** Rice protoplasts were generated from MH63 and harvested for ChIRP assays. The ChIRP assays were conducted using the EZ-Magna ChIRP™ RNA Interactome Kit (Millipore, Bedford, MA, USA). Sonication was performed in ice-cold water bath at setting with high-power, 30 s ON, 30 s OFF pulse intervals for 2 h (total process time). Anti-sense biotinylated DNA tiling probes for selective retrieval of RNA target were designed by online probe design program (www.singlemoleculefish.com), and details are provided in Supplementary Table 1. For *LAIR*, probes were separated into two groups (even and odd), which served as internal control for each other. Probes that targeted *LacZ* mRNA, which absent from rice cell, were used as negative control probes[14].

After ChIRP and RNA/DNA isolation, qRT-PCR was completed using gene-specific primers (Supplementary Data 1) to assess the enrichment of the retrieved RNA and immunoprecipitated DNA, with *OsActin* serving as a negative control gene.

## Data availability

Sequence data in this study have been deposited in the GenBank database and are available with the following accession codes: JX512717, JX512718, JX512719, JX512720, JX512721, JX512722, JX512723, JX512724, JX512725, JX512726, JX512727, and JX512728. The data that support the findings of this study are available from the corresponding authors upon reasonable request.

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

## Acknowledgements

We thank Prof. Aiwu Dong for her valuable discussions and advices, Dr. Bing Liu for his technical assistance with ChIP assay. This work was supported by grants from the National Key Research and Development Program of China (2016YFD0100902), the National Natural Science Foundation of China (31671263), China Postdoctoral Science Foundation Funded Project (KLH1322099), and the Basic Application Research Program from the Shanghai Municipal Agriculture Commission (2014-7-1-2).

## Author contributions

Y.W. designed the study, carried out the experiments, analyzed data, and wrote the paper. X.L. and X.Z. helped with rice plants management in test fields. F.S. performed protoplasts experiments. J.H. helped with the experimental design. W.S. and J.Y. designed the study, analyzed data, and wrote the paper.

## Additional information

**Competing interests:** The authors declare no competing interests.

