## [Peer Review File · Nature Communications]

Reviewers' comments:

Reviewer #1 (Remarks to the Author):

In the manuscript entitled “Long noncoding RNA increases rice grain yield by epigenetically regulating neighboring gene cluster expression”, Wang and colleagues identified an antisense long noncoding RNA (LAIR) in rice, which is involved in regulating expression of genes in the LRK cluster and affects grain yield. In addition, the authors propose that LAIR could associate with potential histone modification complex components and might participate in histone modification status changes at one LRK gene locus. In general, the observations are interesting and there are some correlations between LAIR and LRK gene expression as well as grain yield, however, several necessary experiments to support the mechanistic conclusions drawn are missing.

Major concerns are listed as below:

1. Genetic evidence to support that LAIR affects grain yield through positively regulating LRK cluster gene expression, such as demonstrating that the previously published increase in grain yield in rice overexpressing LRK1 is not reduced when combined with 35S:antiLAIR.
2. The authors assume that the transcriptional regulation of LRK genes by LAIR is direct but no evidence is provided that links LAIR physically to these loci. Since in a heterologous system LAIR overexpression drives increase expression from the LRK4 promoter driving LUC, the association of LAIR with this promoter should be sufficient for the observed effect.
3. Additional concerns are in Figure 3a-c what is the level of LRK gene expression in the WT? The authors should present the raw data (normalized over a housekeeping gene) and note the fold change separately. As presented it is unclear whether any of the observed changes could be biologically relevant. It would be good to also include the LRK1 levels observed in the previously published LRK1 overexpression line that exhibited a change in grain yield as a positive control. It is also important to show the observed spatial expression level of the LRK genes and various LAIR transcripts in the panicle, the leaves, the root, the seeds etc. Are these RNAs expressed in similar tissues? If LAIR helps upregulate the LRKs, is it expressed before these genes in the relevant stages of panicle development?
4. Fig. 4 a begs the question whether histone modification status change (H3K4me3 and

H4K16ac) is the cause or the consequence of LRK gene upregulation in LAIR OE lines. It is well known that altered transcription levels also triggers changes in histone modifications. Given that the promoter of LRK4 is sufficient to see increased expression when co-infiltrated with LAIR, how would this lead to altered histone modification in the gene body under the causal model? Also why did the authors probe LRK1 and not LRK4, which consistently shows the most dramatic response to increased or decreased LAIR levels. Given that anti-LAIR is more physiological, histone modification changes should also be assessed in this genetic background. Finally, since in many cases equivalent levels of histone modifications to the 'activated' state at LRK1 are seen at LRK3 in the WT or in the LAIR overexpression line, there are technical issues or LRK3 is not a good negative control. Since LRK5 is a pseudogene it may be a better negative control.

5. Fig. 4 b-d requires much additional work. The RIP assay was done in vitro protoplast by overexpression of MOF or WDR5 under 35S promoter, which may cause false positives. The proteins tested are RBPs and the critical question is specificity of association. For this heterologous assay – antiLAIR should be included as a negative control. The authors should in addition use the FLAG-tagged versions of WDR5 and MOF they have in rice for in planta RNA-binding assays. In addition, there is no evidence to support that LAIR recruits the histone modification complex (by association with OsMOF, OsWDR5) to the LRK gene loci. Here again the authors should focus on LRK4 and LRK5 and – at a minimum- examine occupancy of OsMOF and OsWDR5 in antiLAIR and WT plants. Combined with more controlled assessment of the specificity of the OsMOF and OsWDR5 proteins with LAIR as described above would provide the much needed support for the model the authors propose.

6. Lastly, the authors should explain in more detail how LAIR was identified and why they chose to focus on MOF/NSL. The changes in H4K16ac are much more subtle than those in H3K4/9/27ac.

Reviewer #2 (Remarks to the Author):

Wang et al reported a long noncoding RNA (lncRNA, called LAIR) that could influence the expression of its neighboring LRK gene cluster. It was interesting that when LAIR was overexpressed, the epigenetic modifications H3K4me3 and H4K16ac were significantly enriched at LRK1, and coincidentally, histone modification proteins OsMOF and OsWDR5 that could bind to the LAIR transcript in a RIP assay, were also enriched at the LRK1 locus. However, I have major concerns that should be solved by the authors. First, I was curious about how LAIR were originally discovered? What were the sequence signature(s) for LAIR to form up to 10 alternative spliced isoforms? What's the functional difference among these splicing isoforms JX512719 to JX512728?

Second, the authors were required to quantify how LAIR regulated grain yield and yield component traits in more details.

Third, did the LAIR and LRK gene cluster influence each other's expression? I assumed that in the transgenic plants down-regulating LAIR expression more histone modifications especially those of transcriptional repressors could be enriched at LRK1 locus. The authors were required to explain what molecular reasons would be for reduced expression of LRKs in anti-LAIR transgenes.

Fourth, the authors were suggested to describe the biological significances for forming the chromatin modifying complexes including OsMOF and/or OsWDR5 in rice plants. Did these genes influence grain yield? How about genetic relationship between these and LAIR or LRKs?

Reviewer #3 (Remarks to the Author):

Long noncoding RNAs have been recognized as important regulators of gene expression in eukaryotes. Up to date, only a couple of long noncoding RNAs have been functionally addressed in planta. Wang et al., in this work provide a very interesting molecular link of the long noncoding RNA LAIR to the improvement of grain yield. The authors report the positive correlation of LAIR expression to the increase of LRKs expression, which may have resulted from the enrichment of H3k4me3 and H4K16ac on LRK loci, leading to the increase of rice yield.

Major comments:

1. In Figure 1 the authors report that compared to the WT, the 35S:LAIR transgenic line shows higher grain yield. Genetically, whether LAIR acts through LRKs to regulate the grain yield is not clear. One can not rule out the possibility that overexpression of the antisense sequence of LAIR (anti-LAIR) may disrupt the promoter of LRK1, possibly other LRKs, and/or produce small RNAs that interfere the expression of LRK1(LRKs). Additional lair knockout or knockdown lines would be necessary for further phenotypic and genetic study. Moreover, pls. provide the expression data of LAIR in anti-LAIR and 35S: LAIR lines by Northern blot.
2. The authors may want to investigate the expression pattern of LAIR during development, and to show whether LAIR is nuclear localized.
3. The authors showed that the expression of all the LRKs are decreased in anti-LAIR lines, however, only the increase of LRK1 and LRK4 expression are detected in 35S: LAIR. Whether

LRK1 and LRK4 are prominent effectors that result in the higher grain yield phenotype in 35S:LAIR? In Figure 3C, the increment of LRK4, LRK6, LRK7 and LRK8 expression are observed in 35S: LAIR-MU1 and 35S:LAIR-MU2, when only LRK4 is significantly increased in 35S: LAIR. Does that mean the expression or activity of LAIR-MU1 and LAIR-MU2 is higher than WT LAIR? In addition, in Figure 3d, whether the P-LRK4 activity is reduced in Anti-LAIR should be examined.

4. The authors demonstrated that LAIR interacts with OsMOF and OsWDR5 in rice by RIP. I'm wondering whether the interaction is direct. Further analysis on this question would be required. More importantly, whether LAIR itself associates with the promoters of LRKs is not known. And whether the load of MOF and WDR5 on the promoter of LRK1 is mediated by LAIR needs to be determined.

Minor concerns:

1. Pls. spell out lncRNA "LAIR" and "LRK" when it appears for the first time in the abstract
2. Line 14, pls. change "transcript" to "is transcribed"
3. Pls. specify which isoform of LAIR is used to construct 35S:LAIR transgenic line in the material and method session

Dear Reviewers:

Thank you for your kind comments on our manuscript entitled “Long noncoding RNA increases rice grain yield by epigenetically regulating neighboring gene cluster expression” (NCOMMS-17-31272). Those comments are all valuable and very helpful for revising and improving our paper, as well as the important guiding significance to our researches. We have studied comments carefully and made correction which we hope meet with approval. The main corrections in the paper and the responds to the reviewer’s comments are listed below:

Responds reviewer’s comments:

Reviewer 1:

In the manuscript entitle “Long noncoding RNA increases rice grain yield by epigenetically regulating neighboring gene cluster expression”, Wang and colleagues identified an antisense long noncoding RNA (LAIR) in rice, which is involved in regulating expression of genes in the LRK cluster and affects grain yield. In addition, the authors propose that LAIR could associate with potential histone modification complex components and might participate in histone modification status changes at one LRK gene locus. In general, the observations are interesting and there are some correlations between LAIR and LRK gene expression as well as grain yield, however, several necessary experiments to support the mechanistic conclusions drawn are missing.

Major concerns are listed as below:

1. Genetic evidence to support that LAIR affects grain yield through positively regulating LRK cluster gene expression, such as demonstrating that the previously published increase in grain yield in rice overexpressing LRK1 is not reduced when combined with 35S:antiLAIR.

Response: Thank you very much for your kind comments. In fact, 35S::anti-LAIR transgenic plants exhibited poor fertility so is inappropriate for the crossing

combination with overexpressing *LRK1* lines. Additionally, the parent materials planting, the hybridization work with the following lines screening will need much longer time in crop rice. Considering the timeliness of the research article, we provide new evidence that antisense *LAIR* (anti-*LAIR*) could suppress the promoter activations of all the *LRKs*, which may be able to be a new support of *LAIR* positively regulating *LRK* cluster gene expression. We now have included the data in Fig. 3e in the revised manuscript.

2. The authors assume that the transcriptional regulation of LRK genes by LAIR is direct but no evidence is provided that links LAIR physically to these loci. Since in a heterologous system LAIR overexpression drives increase expression from the LRK4 promoter driving LUC, the association of LAIR with this promoter should be sufficient for the observed effect.

Response: Thank you very much for your enlightening suggestions. We performed ChIRP (chromatin isolation by RNA purification) to analysis *LRK1* genomic region which was activated by *LAIR*. The result suggested that *LAIR* RNA could physically locate to *LRKs* gene genomic loci, especially on the 5' and 3' untranslated regions of *LRK1*. We now have included the data in Fig. 4f in the revised manuscript.

In this work, we mainly focused on *LRK1*, because our previous studies have revealed the characterization of yield improvement in *LRK1* overexpression lines, and can be the phenotypic reference in *LAIR* studies. The situation of *LRK* genes showed differentiation and complication, and the future study will be aim to characterize the molecular mechanism in other *LRK* genes (such as *LRK4*).

3. Additional concerns are in Figure 3a-c what is the level of LRK gene expression in the WT? The authors should present the raw data (normalized over a housekeeping gene) and note the fold change separately. As presented it is unclear whether any of the observed changes could be biologically relevant. It would be good to also include the LRK1 levels observed in the previously published LRK1 overexpression line that exhibited a change in grain yield as a positive control. It is also important to show the

observed spatial expression level of the LRK genes and various LAIR transcripts in the panicle, the leaves, the root, the seeds etc. Are these RNAs expressed in similar tissues? If LAIR helps unregulated the LRKs, is it expressed before these genes in the relevant stages of panicle development?

Response: Thank you for your kind comments. Figure 3a-c have been changed to Figure 3a,b,d in this revised manuscript, included levels of *LRK* genes expression in WT (shown as black bar). Housekeeping gene *OsActin* was used as a reference control, and gene expression levels in WT lines were used for data normalization (please check Fig. 3). According to your suggestion, we detected expression levels of *LRK1* and *LAIR* observed in the *LRK1* overexpression lines (LRK1-1 and LRK1-2), as shown here. The results suggested that *LRK1* was overexpressed in LRK1-1 and LRK1-2, the expression of *LAIR* was not affected.

To investigate the spatial expression level of *LRK* genes and *LAIR*, qRT-PCR was performed in different tissues (showed in the x-axis), including 3-leaf-stage (Shoot, Seedling Root) and flowering-stage (Flag Leaf, Sheath, Stem, Node, Spikelet, Panicle, Stamen, Pistil, and Caryopsis). As shown below, the expressions of *LRK* genes and *LAIR* in different tissues are different and irregular each other. We can't find any significant correlation or pattern among these RNAs expression levels. The spatial expression of *LAIR* has been supplied in the revised manuscript (please check Line

122-124, and Supplementary Fig. 1).

4. Fig. 4 a begs the question whether histone modification status change (H3K4me3 and H4K16ac) is the cause or the consequence of LRK gene upregulation in LAIR OE lines. It is well known that altered transcription levels also triggers changes in histone modifications. Given that the promoter of LRK4 is sufficient to see increased expression when co-infiltrated with LAIR, how would this lead to altered histone modification in the gene body under the causal model? Also why did the authors probe LRK1 and not LRK4, which consistently shows the most dramatic response to increased or decreased LAIR levels. Given that anti-LAIR is more physiological, histone modification changes should also be assessed in this genetic background. Finally, since in many cases equivalent levels of histone modifications to the ‘activated’ state at LRK1 are seen at LRK3 in the WT or in the LAIR overexpression line, there are technical issues or LRK3 is not a good negative control. Since LRK5 is a pseudogene it may be a better negative control.

Response: Thank you very much for your suggestions. Recent studies have reported that enrichments of H3K4me3 and H4K16ac are important for activating transcription^{1, 2}. In our work, the histone modification was increased in *LRK1* upregulated 35S::LAIR lines (Fig. 4b), and reduced in *LRK1* downregulated 35S::anti-LAIR lines (Fig. 4c), indicated it is possible that enrichments of histone

modification status promote the expression of *LRK* gene. The regulation of gene activities is usually mutual, not unilateral. There may also exist the possibility of changes of histone modification status resulted from *LRK* gene. We agree with your profound consideration, and will seriously consider this question in our future work.

(References: 1. Fromm M, Avramova Z. ATX1/AtCOMPASS and the H3K4me3 marks: how do they activate *Arabidopsis* genes? *Curr Opin Plant Biol.* **21**, 75-8 (2014); 2. Conrad T *et al.* The MOF chromobarrel domain controls genome-wide H4K16 acetylation and spreading of the MSL complex. *Dev Cell.* **22**,610-624 (2012))

As described above, the situation of *LRKs* gene showed differentiation and complication, so we mainly focused on *LRK1* in this work, because some working foundations such as *LRK1* overexpression lines have been studied previously, and can be the phenotypic reference in *LAIR* studies. The future studies will be aim to characterize the histone modification in other *LRK* genes (such as *LRK4*).

Thank you very much for your enlightening question. According to your suggestions, the histone modification was analyzed in 35S::anti-*LAIR* lines, reduction were observed both at the *LRK1* and *LRK3* locus (Fig. 4c), which were consistent with the suppression of anti-*LAIR* to *LRKs* promoter (Fig. 3d, e). We now have included the new ChIP data using 35S::anti-*LAIR* lines in the revised manuscript (please check Fig. 4c).

Thank you for your suggestions. We used *LRK3* as negative control, because its transcription was unaffected by *LAIR* overexpression (Fig. 3a-c), and our ChIP-qRT-PCR results indicated that H3K4me3 and H4K16ac were unaffected at the *LRK3* locus, which correlated with the gene expression levels. *LRK5* is not expressing in MH63, no expression data can be offered for correlation analysis, so *LRK5* is not a suitable control.

5. Fig. 4 b-d requires much additional work. The RIP assay was done in vitro protoplast by overexpression of MOF or WDR5 under 35S promoter, which may cause false positives. The proteins tested are RBPs and the critical question is specificity of association. For this heterologous assay – antiLAIR should be included

as a negative control. The authors should in addition use the FLAG-tagged versions of WDR5 and MOF they have in rice for in planta RNA-binding assays. In addition, there is no evidence to support that LAIR recruits the histone modification complex (by association with OsMOF, OsWDR5) to the LRK gene loci. Here again the authors should focus on LRK4 and LRK5 and – at a minimum- examine occupancy of OsMOF and OsWDR5 in antiLAIR and WT plants. Combined with more controlled assessment of the specificity of the OsMOF and OsWDR5 proteins with LAIR as described above would provide the much needed support for the model the authors propose.

Response: Thank you very much for your suggestions. In our RIP analysis, three proteins were recruited and expressed, respectively. The results showed significant binding of OsMOF and OsWDR5, but not OsMLE. So OsMLE might be able to be a protein negative control to avoid false positives. According to your suggestions, 35S::anti-LAIR has been included into RIP as a negative control, which showed no binding signal in all three proteins (please check Fig. 4d). Moreover, these works were further confirmed by RIP assay using the OsMOF-FLAG and OsWDR5-FLAG transgenic rice lines, the results showed significant binding of OsMOF and OsWDR5 with *LAIR* (please check Supplementary Fig. 4). We now have included the new RIP data in the revised manuscript (please check revised Fig. 4d, and Supplementary Fig. 4).

According to your suggestion, we performed ChIRP to analysis interaction of *LAIR* with *LRK* genomic region, the result suggested that *LAIR* RNA could physically locate to the 5' and 3' untranslated regions of *LRK1* gene loci. In this revised manuscript, we provided evidences that *LAIR* interacted with chromatin-modifying proteins (by RIP, Fig. 4d), chromatin-modifying proteins enriched at the *LRK1* genomic region (by ChIP, Fig. 4e), and *LAIR* RNA could physically locate to *LRK1* gene genomic DNA region (by ChIRP, Fig. 4f), so we proposed the model that *LAIR* might help recruit chromatin-modifying proteins to *LRK1* genomic region to activate transcription in discussion in the revised manuscript (please check Line 364-373).

As described above, we mainly focused on *LRK1* in this work, because some

working foundations such as *LRK1* overexpression lines have been studied previously. The characterization of other LRK genes (such as *LRK4*) will be studied in the future work. *LRK5* is not a suitable control, because *LRK5* is not expressing in MH63, no expression data can be offered for correlation analysis.

Thank you very much for your suggestions on the control of OsMOF and OsWDR5 combine with *LAIR*. 35S::anti-*LAIR* has been included into RIP as a negative control, which showed no binding signal in all three proteins (Fig. 4d). OsMOF-FLAG and OsWDR5-FLAG transgenic rice lines were used in RIP, the results showed significant binding of OsMOF and OsWDR5 with *LAIR* (Supplementary Fig. 4), combine with chromatin-modifying proteins coordinate the enrichment of H4K16Ac and H3K4me3 in the *LRK1* genomic region (Fig. 4b, e), and *LAIR* RNA could physically locate to *LRK1* gene genomic DNA region (Fig. 4f), to provide the support for the model we propose in discussion.

6. Lastly, the authors should explain in more detail how *LAIR* was identified and why they chose to focus on MOF/NSL. The changes in H4K16ac are much more subtle than those in H3K4/9/27ac.

Response: Thank you very much for your comments. The identified of *LAIR* from our previously identification of a QTL for improved rice yield, which consisted of several tandemly arranged intronless *LRK* gene cluster. Transcriptomic analyses revealed an antisense transcript encoded by the 5' terminal region of the *LRK* gene cluster, which was named *LAIR*. That information has been described in the revised manuscript in Line 109-117, please check.

Because of the histone modification changes of H3K4me3 and H4K16ac in ChIP assay (Fig, 4b, c), we chose to focus on MOF (the primary histone H4K16 acetyltransferase)/ NSL (MOF is reportedly a component of NSL). The changes in H4K16ac might be little subtle in *LRK1*, but were unaffected at the *LRK3* locus, which were consistent with the gene transcription levels in the 35S::*LAIR* lines. But H3K4/9/27ac showed no consistent at the *LRK3* locus. Here again, thank you very much for your suggestions on 35S::anti-*LAIR*, ChIP assay of 35S::anti-*LAIR* helped

us confirmed the H3K4me3 and H4K16ac changes in *LAIR* associated molecular process. The new ChIP data using 35S::anti-LAIR lines have been included in the revised manuscript (please check Fig. 4c).

Reviewer 2:

Wang et al reported a long noncoding RNA (lncRNA, called LAIR) that could influence the expression of its neighboring LRK gene cluster. It was interesting that when LAIR was overexpressed, the epigenetic modifications H3K4me3 and H4K16ac were significantly enriched at LRK1, and coincidentally, histone modification proteins OsMOF and OsWDR5 that could bind to the LAIR transcript in a RIP assay, were also enriched at the LRK1 locus. However, I have major concerns that should be solved by the authors. **First, I was curious about how LAIR were originally discovered? What were the sequence signature(s) for LAIR to form up to 10 alternative spliced isoforms? What's the functional difference among these splicing isoforms JX512719 to JX512728?**

Response: Thank you very much for your comments. The identified of *LAIR* from our previously identification of a QTL for improved rice yield, which consisted of several tandemly arranged intronless *LRK* gene cluster. Transcriptomic analyses revealed an antisense transcript encoded by the 5' terminal region of the *LRK* gene cluster, which was named *LAIR*. That information has been described in the revised manuscript in

WT LAIR LAIR-short

Line 109-117, please check.

The 10 alternative spliced isoforms of *LAIR* were a very interesting feature, on the one hand different alternatively isoforms had diverse coding frames which proofed *LAIR* as noncoding RNA, on the other hand different alternatively isoforms might involve in the different regulatory effects among the *LRK* alleles. According to our currently works, splicing isoforms JX512726 (*LAIR*) and JX512719 (*LAIR*-short) both showed similar morphological characteristics and grain yield increase ability. The isoforms of *LAIR* and *LAIR*-short is showed above, with the phenotypes of their overexpressing transgenic rice plants, please check.

Second, the authors were required to quantify how *LAIR* regulated grain yield and yield component traits in more details.

Response: Thank you very much for your suggestions. More details of yield component might need new generation and much longer time in crop rice, the data cannot be provide right now.

Third, did the *LAIR* and *LRK* gene cluster influence each other's expression? I assumed that in the transgenic plants down-regulating *LAIR* expression more histone modifications especially those of transcriptional repressors could be enriched at *LRK1* locus. The authors were required to explain what molecular reasons would be for reduced expression of *LRKs* in anti-*LAIR* transgenes.

Response: Thank you very much for your suggestions. We checked the expression level of *LAIR* in the *LRK1* overexpression lines (*LRK1*-1 and *LRK1*-2), which suggested that overexpressed *LRK1* did not influence *LAIR* expression level (please check the Figure showed below), but in the *LAIR*-overexpressing lines *LRK1* transcript levels was high (Fig. 3a) and *LRK1* were expressed at low levels in the 35S::anti-*LAIR* lines (Fig. 3d). The results suggested that *LAIR* could influence *LRK* gene, but *LRK1* could not influence *LAIR*.

Thank you very much for your enlightening question. According to your suggestions, the histone modification was analyzed in 35S::anti-*LAIR* lines, reduction

were observed both at the *LRK1* and *LRK3* locus (Fig. 4c), which were consistent with the suppression of anti-LAIR to *LRKs* promoter (Fig. 3d, e). According to these results, molecular reasons for the reduced expression of *LRKs* in anti-LAIR transgenes, may be caused by the reduction of histone modifications associated with transcriptional activation (H3K4me3 and H4K16ac) at the *LRK* locus. Thank you for your suggestion, more histone modifications especially those of transcriptional repressors would be investigated in our future work. We now have included the new ChIP data using 35S::anti-LAIR lines in the revised manuscript (please check Fig. 4c).

Fourth, the authors were suggested to describe the biological significances for forming the chromatin modifying complexes including OsMOF and/or OsWDR5 in rice plants. Did these genes influence grain yield? How about genetic relationship between these and LAIR or LRKs?

Response: Thank you very much for your suggestions. Although MOF has been well studied in human and fruit fly, little is known about the function of rice homolog of human MOF in rice. Recently, there are no published reports describing lncRNA-regulated gene expression associated with the histone modification protein OsMOF in rice. The NSL complex is a universal and sex-independent major regulator of housekeeping genes. Based on interactions between *LAIR* and OsMOF/ OsWDR5,

we speculated that *LAIR* functions are associated with the evolutionarily conserved NSL complex. These biological significances of OsMOF and OsWDR5 in rice plants have been described in discussion in the revised manuscript (please check Line 375-382).

According to our currently works, the morphological characteristics of OsMOF-FLAG and OsWDR5-FLAG transgenic lines showed diverse features, which might be caused by the extensive functions of *MOF* and *WDR5* gene. In our proposed model, *LAIR* increasing rice grain yield by recruiting chromatin-modifying proteins to the neighbouring *LRK* gene cluster to activate transcription in *trans*. The grain yield phenotype mainly came from functions of *LRK* gene cluster, OsMOF and OsWDR5 might assist *LAIR* to promote *LRK* gene expression by introducing active histone modifications to specific *LRK* genes loci (Fig. 4g).

Reviewer 3:

Long noncoding RNAs have been recognized as important regulators of gene expression in eukaryotes. Up to date, only a couple of long noncoding RNAs have been functionally addressed in plants. Wang et al., in this work provide a very interesting molecular link of the long noncoding RNA LAIR to the improvement of grain yield. The authors report the positive correlation of LAIR expression to the increase of LRKs expression, which may have resulted from the enrichment of H3k4me3 and H4K16ac on LRK loci, leading to the increase of rice yield.

Major comments:

1. In Figure 1 the authors report that compared to the WT, the 35S:LAIR transgenic line shows higher grain yield. Genetically, whether LAIR acts through LRKs to regulate the grain yield is not clear. One can not rule out the possibility that overexpression of the antisense sequence of LAIR (anti-LAIR) may disrupt the promoter of LRK1, possibly other LRKs, and/or produce small RNAs that interfere the expression of LRK1(LRKs). Additional lair knockout or knockdown lines would be necessary for further phenotypic and genetic study. Moreover, pls. provide the expression data of LAIR in anti-LAIR and 35S: LAIR lines by Northern blot.

Response: Thank you very much for your suggestions. We checked the expression level of *LAIR* in the *LRK1* overexpression lines (LRK1-1 and LRK1-2), which suggested that overexpressed *LRK1* did not influence *LAIR* expression level (please check the Figure showed here), but in the *LAIR*-overexpressing lines *LRK1* transcript

levels was high (Fig. 3a) and *LRK1* were expressed at low levels in the 35S::anti-LAIR lines (Fig. 3d). The results suggested that *LAIR* could influence *LRK* gene, but *LRK1* could not influence *LAIR*. Additionally, the 35S::LAIR transgenic plants exhibited similar growth traits with *LRK1*-overexpressing transgenic lines. Therefore, *LAIR* might affect rice plant growth and grain yield by regulating the expression of *LRK* genes.

Another anti-LAIR transgenic line genetic transformation (35S::anti-LAIR-3') has been conducted at same time with 35S::anti-LAIR which already showed in our manuscript, 35S::anti-LAIR-3' showed similar exhibited inhibited growth and decreased grain yield with 35S::anti-LAIR lines (Fig. 1b). The antisense *LAIR* region used in 35S::anti-LAIR and 35S::anti-LAIR-3' vector were showed below. These two regions separated on different 5' and 3' untranslated regions of *LRK1*, which could not both disrupt the promoter of *LRK1*, and showed the similar morphological characteristics, which could be supplementary proof of *LAIR* function. For the very close combining of *LAIR* and *LRK1* in chromosome loci, *LAIR* knockout might influent the genome structure and regular expression of *LRK1*.

According to your suggestion, we performed Northern blot to explore the expression of *LAIR* in 35S::anti-*LAIR* and 35S::*LAIR* lines for several times, but no signal was detected. We analyzed possible reasons. Firstly, because the expression level of *LAIR* in all lines is not rich enough, it is difficult to detect by Northern blot. Secondly, there exist many different alternatively spliced *LAIR* isoforms, so the predicted Northern blot result should appear as a smear, which will greatly increase the difficulty to get *LAIR* band signals. So in this study we considered qRT-PCR as a suitable method to analyze the expression of *LAIR*.

2. The authors may want to investigate the expression pattern of *LAIR* during development, and to show whether *LAIR* is nuclear localized.

Response: Thank you very much for your suggestions. The spatial and temporal expression of *LAIR* was analysed, *LAIR* showed relatively high levels in 3-leaf-stage shoot, flowering-stage node, pistil and caryopsis. This information has been supplied in the revised manuscript (please check Line 122-124, and Supplementary Fig. 1).

We performed ChIRP (chromatin isolation by RNA purification) in the revised manuscript, and the result suggested that *LAIR* RNA could physically locate to *LRKs*

gene genomic loci, which meant that *LAIR* existed in nuclear (please check Fig. 4f).

3. The authors showed that the expression of all the LRKs are decreased in anti-LAIR lines, however, only the increase of LRK1 and LRK4 expression are detected in 35S: LAIR. Whether LRK1 and LRK4 are prominent effectors that result in the higher grain yield phenotype in 35S: LAIR? In Figure 3C, the increment of LRK4, LRK6, LRK7 and LRK8 expression are observed in 35S: LAIR-MU1 and 35S:LAIR-MU2, when only LRK4 is significantly increased in 35S: LAIR. Does that mean the expression or activity of LAIR-MU1 and LAIR-MU2 is higher than WT LAIR? In addition, in Figure 3d, whether the P-LRK4 activity is reduced in Anti-LAIR should be examined.

Response: Thank you very much for your comments. The different regulatory effects between sense *LAIR* and antisense *LAIR*, may be caused by the fact that sense *LAIR* transformation only contained one isoform (JX512726) of alternative spliced *LAIR*, and antisense *LAIR* could interfere all isoforms. So, we can only speculate *LRK1* and *LRK4* are prominent effectors that result in the higher grain yield phenotype in JX512726-35S::LAIR lines. This explanation has been supplied in the revised manuscript (please check Line 225-228).

We found the interesting activity of LAIR-MU too, and supposed that might because the mutation influenced the molecular structure of *LAIR* RNA, which was important for the lncRNA function.

Thank you very much for your suggestions. The promoter activations of all the *LRKs* has been examined by co-infiltrated with antisense *LAIR* (anti-LAIR). The result showed that anti-LAIR suppressed the promoter activations of all the *LRKs*, including P-LRK4. This information has been supplied in the revised manuscript (please check Fig. 3e).

4. The authors demonstrated that LAIR interacts with OsMOF and OsWDR5 in rice by RIP. I'm wondering whether the interaction is direct. Further analysis on this question would be required. More importantly, whether LAIR itself associates with

the promoters of LRKs is not known. And whether the load of MOF and WDR5 on the promoter of LRK1 is mediated by LAIR needs to be determined.

Response: Thank you very much for your suggestions. We understand that the direct interaction between *LAIR* and OsMOF/ OsWDR5 may better reveal their interaction relationship. According to the results, we can only conclude that *LAIR* could interact with OsMOF and OsWDR5 in rice. Whether the interaction is direct is not clear, there is possibility that *LAIR* interact with OsMOF/ OsWDR5 through epigenetic modification complex (such as NSL complex). However, in the present study, we mainly focused on the regulation of *LAIR* to accomplish the histone modification on *LRK* loci, the evidence may be cannot proof the direct or indirect interaction, but should be sufficient to draw a conclusion that OsMOF and OsWDR5 are involved in the regulatory process of *LAIR*.

Thank you very much for your enlightening suggestions. We performed ChIRP (chromatin isolation by RNA purification) to analysis *LRK1* genomic region which was activated by *LAIR*. The result suggested that *LAIR* RNA could physically locate to *LRKs* gene genomic loci, especially on the 5' and 3' untranslated regions of *LRK1*. We now have included the data in Fig. 4f in the revised manuscript.

Finally, we provided evidences that *LAIR* interacted with chromatin-modifying proteins (Fig. 4d), chromatin-modifying proteins enriched at the *LRK1* genomic region (Fig. 4e), and *LAIR* RNA could physically locate to *LRK1* gene genomic DNA region (Fig. 4f). In this revised manuscript, CHIP assay of histone modifications using 35S::anti-*LAIR* lines were supplied, reduction of H3K4me3 and H4K16ac were observed at the *LRK1* locus (Fig. 4c), which consistent with the suppression of anti-*LAIR* to *LRKs* (Fig. 3d, e). This result suggested that knockdown of *LAIR* weakened H4K16Ac and H3K4me3 occupations in the *LRK1* genomic region, combining with the enrichment of H3K4me3 and H4K16ac on *LRK1* loci in 35S::*LAIR* lines (Fig. 4b), we proposed chromatin-modifying proteins functioned via changing abundance of histone modifications. The result implied that it is possible that *LAIR* mediated the load of OsMOF and OsWDR5 on the promoter of *LRK1*.

Minor concerns:

1. Pls. spell out lncRNA “LAIR” and “LRK” when it appears for the first time in the abstract

Response: We very much appreciate your careful reading of our manuscript. The full name has been added in abstract in the revised manuscript, please check Line 28 and Line 29.

2. Line 14, pls. change “transcript” to “is transcribed”

Response: We are very sorry for our incorrect writing. The change has been made, and in this revised version, we have checked carefully through the whole manuscript and corrected the mistakes been found.

3. Pls. specify which isoform of LAIR is used to construct 35S:LAIR transgenic line in the material and method session

Response: Thank you very much for your suggestions. This information has been supplied in the methods session in the revised manuscript, please check Line 411-413.

We tried our best to improve the manuscript and made some changes in the manuscript. We appreciate for Editors/Reviewers’ warm work earnestly, and hope that the correction will meet with approval.

Once again, thank you very much for your comments and suggestions.

Reviewers' Comments:

Reviewer #1 (Remarks to the Author):

The revised manuscript is much improved. The authors have addressed my concerns satisfactorily.

Reviewer #2 (Remarks to the Author):

The authors have clearly solved all of my concerns.

Reviewer #3 (Remarks to the Author):

The authors have tried the best to answer my previous concerns. This work shows the interesting correlation of the expression a long noncoding RNA LAIR with the improvement of grain yield. However, I think the data for the mechanistic conclusions are still not evident.

Major concerns:

1. The genetic evidence on whether LAIR acts through LRKs to regulate the grain yield should be provided. For example, whether the loss of LRK1 and/or LRK4 in 35S:LAIR would reduce the grain yield?
2. The authors suggest that LAIR increases the H3K4me3 and H4K16ac at LRK loci by associating with OsMOF and OsWDR5. I would suggest testing the association of those active histone marks at LRK4 locus. Since the changes of H3K4me3 and H4K16ac levels at LRK1 locus in 35:LAIR and anti-LAIR lines may be caused by the transcription of LAIR itself. Moreover, whether the LAIR can mediate the recognition of OsMOF and OsWDR5 to its target LRKs is not addressed in the current version.

Dear Reviewers:

Thank you for your kind comments on our manuscript entitled “Long noncoding RNA increases rice grain yield by epigenetically regulating neighboring gene cluster expression” (NCOMMS-17-31272A). Those comments are important guiding to our further researches, and very valuable for revising and details accuracy improving of our paper. We have studied comments carefully and made correction which we hope meet with approval. **Revised portion have been highlighted in red in the revised manuscript and revised supplementary information.** The main corrections in the paper and the responds to the reviewer’s comments are listed below:

Responds reviewer’s comments:

Reviewer 1:

The revised manuscript is much improved. The authors have addressed my concerns satisfactorily.

Reviewer 2:

The authors have clearly solved all of my concerns.

Reviewer 3:

The authors have tried the best to answer my previous concerns. This work shows the interesting correlation of the expression a long noncoding RNA LAIR with the improvement of grain yield. However, I think the data for the mechanistic conclusions are still not evident.

Major concerns:

1. The genetic evidence on whether LAIR acts through LRKs to regulate the grain yield should be provided. For example, whether the loss of LRK1 and/or LRK4 in 35S:LAIR would reduce the grain yield?

Response: Thank you very much for your enlightening suggestions. We totally understand that the loss of *LRK1* and/or *LRK4* in 35S:LAIR may better give genetic evidences to support that *LAIR* acts through *LRKs* to regulate the grain yield. Actually, at very beginning of this research, we have planned to generate the loss-of-function mutant of *LRK1* lines, but there exist many difficulties. Firstly, the physical location of *LAIR* and *LRK1* was very close (Fig. 1a), knockout of *LRK1* promoter may disrupt the sequence and transcription of original *LAIR*. Secondly, the coding sequences of all members of *LRK* gene cluster had no intron, and showed highly conserved sequence identity with each other, which made it difficult to knock out a single gene specifically via coding sequences. Moreover, the phenotype of lines that loss of *LRK1* and/or *LRK4* may not be significant, as the result of gene redundancy of *LRK* gene cluster. Finally, we fully agree that your suggestions are very worth to try, but a series of lines that loss of *LRK1* and/or *LRK4* in 35S:LAIR need be generated, it will need time to found the appropriate lines which had highly specific, including the homozygous lines screening and the phenotype analysis, one-two years will be needed in rice. Considering the timeliness of the research article, we will seriously take your advices and consider the evidences in the future studies on the relationship between *LAIR* alternative spliced isoforms and different *LRK* cluster genes.

2. The authors suggest that LAIR increases the H3K4me3 and H4K16ac at LRK loci by associating with OsMOF and OsWDR5. I would suggest testing the association of those active histone marks at LRK4 locus. Since the changes of H3K4me3 and H4K16ac levels at LRK1 locus in 35:LAIR and anti-LAIR lines may be caused by the transcription of LAIR itself. Moreover, whether the LAIR can mediate the recognition of OsMOF and OsWDR5 to its target LRKs is not addressed in the current version.

Response: Thank you very much for your carefully analysis on our manuscript. According to your suggestion, we immediately tested levels of histone modifications on *LRK4* locus, the results showed that H3K4me3 and H4K16ac levels were increased in the 35S::LAIR lines, and reduced in the 35S::anti-LAIR lines. We now have included the data in Supplementary Fig. 5 (please check Revised Supplementary

information Line 13, 50-59), and supplied the results in the revised manuscript (please check Line 212-217). The qRT-PCRs were performed with the same immunoprecipitated chromatin fragments ever used in Fig. 4b and c, which were reserved in -80°C. Gene-specific primers of *LRK4* were listed in Supplementary Table 1 (please check Revised Supplementary information Line 61). Here again, thank you very much for your suggestion, which helped us confirmed the H3K4me3 and H4K16ac changes in *LAIR* associated molecular process, and made the results and conclusions more precise.

Thank you very much for your question. We understand your concerns about whether the *LAIR* can mediate the recognition of OsMOF and OsWDR5 to its target *LRKs*. According to the results, we only concluded that *LAIR* could interact with OsMOF and OsWDR5 in rice, both lncRNA and epigenetic modification proteins located to specific *LRK* genes to promote gene expression by introducing active histone modifications (Fig. 4). Whether the recognition is mediated by *LAIR* is not clear. So we have carefully checked the whole manuscript, and corrected the description of our model of *LAIR* during the regulation of *LRKs* (please check Line 306-307, 682) in this revised manuscript. We very much appreciate your careful reading and accurate correction of our molecular model.

Other changes for format requirements (Revised manuscript):

1. Introduction section mark: Line 40.
2. Remove Figures inserted in manuscript text before: Line 96, 137, 163, and 220.
3. Discussion subheadings deletion: Line 265, 289, and 319.
4. Data availability: Line 459-460.
5. Competing interests: Line 604-605.

We tried our best to improve the manuscript and made appropriate changes in the manuscript. We appreciate for Editors/Reviewers' warm work earnestly, and hope that the correction will meet with approval.

Once again, thank you very much for your comments and suggestions.

Reviewers' Comments:

Reviewer #3 (Remarks to the Author):

The authors have addressed the issues raised in the last version to some extent. It's very interesting to see the over-expression of an lncRNA, LAIR, is positively correlated with the grain yield. However, the important genetic data on whether the LAIR acts through LRKs to regulate the grain yield is still lacking. In addition, the results in the current version have not provided solid evidence to give the picture of precise molecular mechanism of LAIR function in rice. For instance, the authors only showed that the histone marks of active transcription H3K4me3 and H4K16ac were increased in 35S:LAIR lines at LRK1 and LRK4, but whether LAIR associates with both loci and mediates the loading of the two histone marks to the target sites by directly interacting with OsMOF and OsWDR5 require further investigation. I believe the addition of those information mentioned above to the current version will make this manuscript more suitable for publication in Nature Communications.

Dear Reviewer:

Thank you for your kind comments on our manuscript entitled “Long noncoding RNA increases rice grain yield by epigenetically regulating neighboring gene cluster expression” (NCOMMS-17-31272B). Those comments are all valuable and very helpful for the revising and improving of our manuscript, also the important guiding to our further researches. We have studied the comments carefully and made correction which we hope will meet with approval. Revised manuscript including both ClearCopy and TrackedCopy (in which all changes are tracked) are submitted. All **Line numbers** are mentioned according to the position of TrackedCopy. The responds to the reviewer’s comments are listed below:

Reviewer’s comments:

Reviewer #3:

The authors have addressed the issues raised in the last version to some extent. It’s very interesting to see the over-expression of an lncRNA, LAIR, is positively correlated with the grain yield. However, the important genetic data on whether the LAIR acts through LRKs to regulate the grain yield is still lacking. In addition, the results in the current version have not provided solid evidence to give the picture of precise molecular mechanism of LAIR function in rice. For instance, the authors only showed that the histone marks of active transcription H3K4me3 and H4K16ac were increased in 35S:LAIR lines at LRK1 and LRK4, but whether LAIR associates with both loci and mediates the loading of the two histone marks to the target sites by directly interacting with OsMOF and OsWDR5 require further investigation. I believe the addition of those information mentioned above to the current version will make this manuscript more suitable for publication in Nature Communications.

Response: Thank you very much for your enlightening suggestions. We totally

understand your concern on whether the *LAIR* acts through *LRKs* to regulate the grain yield is still not clear. Thus, in this revised version of our manuscript, all inappropriate statements about links between *LAIR* and yield via *LRKs* have been modified throughout the text (please check Line 28-33 and 152-155), which are described as *LAIR* overexpression increases grain yield and regulates neighboring gene cluster expression. Moreover, the title of the manuscript has been changed to "A long noncoding RNA overexpression increases rice grain yield and regulates neighbouring gene cluster expression" (please check Line 2-4).

In addition, we agree with you that it isn't clear whether *LAIR* associates with both loci and mediates the loading of the two histone marks, and whether the changes in histone marks are the cause of *LRK* expression or occurring as a consequence. Thus, we remove statements like "*LAIR* induces the epigenetic modification of *LRK* genes by interacting with histone modification proteins at specific *LRK* gene loci". In this revised manuscript, for accurate description of our current conclusion, we change to conclude that *LAIR* overexpression results in altered expression and changes in histone marks. Please check Line 98-99, 200-201, 205-206, 227-231, 261-263, 308-310, 322-324, 696-698 and 727-728. In this study, we identified the cooperation involving *LAIR* RNA, epigenetic modification proteins and *LRK* gene genome DNA. We understand that the molecular mechanism of *LAIR* function is not very precisely. Given that the current understanding on the mechanisms underlying lncRNAs function in crop species remains limited compare with other species, we believe our study is helpful for enhancing the understanding and exploring research methods on lncRNAs in crop, and wish to share those information for communication and inspiration with other works on lncRNAs. Thank you very much for your suggestions, whether *LAIR* associates with both loci and mediates the loading of the two histone marks to the target sites by directly interacting with OsMOF and OsWDR5, will be investigated in our further study.

We tried our best to improve the manuscript and made appropriate changes in the

revised manuscript. We appreciate for Editors/Reviewers' warm work earnestly, and hope that the correction will meet with approval.

Once again, thank you very much for your comments and suggestions.